# Combining Viral Genetics and Statistical Modeling to Improve HIV-1 Time-of-Infection Estimation towards Enhanced Vaccine Efficacy Assessment

**DOI:** 10.3390/v11070607

**Published:** 2019-07-03

**Authors:** Raabya Rossenkhan, Morgane Rolland, Jan P.L. Labuschagne, Roux-Cil Ferreira, Craig A. Magaret, Lindsay N. Carpp, Frederick A. Matsen IV, Yunda Huang, Erika E. Rudnicki, Yuanyuan Zhang, Nonkululeko Ndabambi, Murray Logan, Ted Holzman, Melissa-Rose Abrahams, Colin Anthony, Sodsai Tovanabutra, Christopher Warth, Gordon Botha, David Matten, Sorachai Nitayaphan, Hannah Kibuuka, Fred K. Sawe, Denis Chopera, Leigh Anne Eller, Simon Travers, Merlin L. Robb, Carolyn Williamson, Peter B. Gilbert, Paul T. Edlefsen

**Affiliations:** 1Vaccine and Infectious Disease Division, Fred Hutchinson Cancer Research Center, Seattle, WA 98109, USA; 2U.S. Military HIV Research Program, Walter Reed Army Institute of Research, Silver Spring, MD 20910, USA; 3Henry M. Jackson Foundation for the Advancement of Military Medicine, Inc., Bethesda, MD 20817, USA; 4South African Medical Research Council Bioinformatics Unit, South African National Bioinformatics Institute, University of the Western Cape, Cape Town 7535, South Africa; 5Department of Pathology, Faculty of Health Sciences, Institute of Infectious Disease and Molecular Medicine, University of Cape Town, Cape Town 7925, South Africa; 6Armed Forces Research Institute of Medical Sciences, Bangkok 10400, Thailand; 7Makerere University Walter Reed Project, Kampala, Uganda; 8Kenya Medical Research Institute/U.S. Army Medical Research Directorate-Africa/Kenya-Henry Jackson Foundation MRI, Kericho 20200, Kenya; 9Sub-Saharan African Network for TB/HIV Research Excellence (SANTHE), Africa Health Research Institute, Durban 4001, South Africa; 10Department of Biostatistics, University of Washington, Seattle, WA 98195, USA

**Keywords:** HIV-1, infection time, founder multiplicity, leave-one-out-cross-validation (LOOCV), HIV-1 primary infection, sequence analysis, vaccine efficacy assessment, acute and early HIV-1 infection

## Abstract

Knowledge of the time of HIV-1 infection and the multiplicity of viruses that establish HIV-1 infection is crucial for the in-depth analysis of clinical prevention efficacy trial outcomes. Better estimation methods would improve the ability to characterize immunological and genetic sequence correlates of efficacy within preventive efficacy trials of HIV-1 vaccines and monoclonal antibodies. We developed new methods for infection timing and multiplicity estimation using maximum likelihood estimators that shift and scale (calibrate) estimates by fitting true infection times and founder virus multiplicities to a linear regression model with independent variables defined by data on HIV-1 sequences, viral load, diagnostics, and sequence alignment statistics. Using Poisson models of measured mutation counts and phylogenetic trees, we analyzed longitudinal HIV-1 sequence data together with diagnostic and viral load data from the RV217 and CAPRISA 002 acute HIV-1 infection cohort studies. We used leave-one-out cross validation to evaluate the prediction error of these calibrated estimators versus that of existing estimators and found that both infection time and founder multiplicity can be estimated with improved accuracy and precision by calibration. Calibration considerably improved all estimators of time since HIV-1 infection, in terms of reducing bias to near zero and reducing root mean squared error (RMSE) to 5–10 days for sequences collected 1–2 months after infection. The calibration of multiplicity assessments yielded strong improvements with accurate predictions (ROC-AUC above 0.85) in all cases. These results have not yet been validated on external data, and the best-fitting models are likely to be less robust than simpler models to variation in sequencing conditions. For all evaluated models, these results demonstrate the value of calibration for improved estimation of founder multiplicity and of time since HIV-1 infection.

## 1. Introduction

Accurately estimating the date of HIV-1 infection from samples collected during early infection is important for epidemiology and molecular virology studies—it is also critically important for the in-depth analysis of clinical trials to evaluate the efficacy of various infection prevention modalities including HIV-1 vaccines and monoclonal antibodies (mAbs). During the acute phase of HIV-1 infection, infected individuals progress through a series of six classifiable stages (the “Fiebig stages”) [1,2,3,4]. Progression through these stages is often used to estimate the time of HIV-1 infection in clinical and epidemiological studies [5,6,7,8]. However, for the purposes of some studies, the frequency at which HIV-1 diagnostic testing of uninfected individuals would need to be conducted to achieve sufficient accuracy and precision of this estimate makes this approach intractable. As an alternative, approaches leveraging serological/immunological biomarkers, multi-assay algorithms, or HIV-1 viral sequence analysis (reviewed in [9]) have been developed, many of which perform well when analyzing data from only a single time point. For instance, sequence-based “molecular clock” approaches leverage the observed approximately linear increase in HIV-1 sequence diversity over time in an infected individual [10,11,12,13,14,15,16,17,18]. One such approach utilizing next generation sequencing data has been shown to have improved accuracy and precision compared to Sanger sequencing-based methods, and to have good performance even when the sequencing data are obtained several years after infection [14].

However, despite the many developments in estimators of HIV-1 infection timing for other applications, we are motivated by the need for greater accuracy and precision than current methods can achieve for a specific application: estimating infection dates in clinical efficacy trials that have primary endpoint HIV-1 infection, which is diagnosed based on HIV tests administered at a series of longitudinal study visits. These estimated dates are used for (1) the study of time-dependent marker correlates of risk (CoR) of HIV-1 infection in HIV-1 vaccine efficacy (VE) trials and mAb prevention efficacy trials, and (2) “sieve analysis” of how the level of vaccine/prevention efficacy depends on genotypic characteristics of HIV-1 at acquisition in these trials, as summarized below. The focus of our study is to evaluate and optimize day-of infection estimators using data available from a single sample collected in a time frame reflecting diagnosis intervals typical of HIV-1 prevention efficacy trials: 1–2 months, or approximately six months post infection. These data include aligned HIV-1 nucleotide sequences and plasma viral load from the first HIV-positive sample; in addition, the dates and results of HIV-1 diagnostic tests for at least the last negative and first positive diagnosis dates available for all infected participants. The defining characteristic of this data setup is a prospective cohort study of HIV-1 negative individuals with frequent and periodic testing for HIV-1 infection.

In HIV-1 vaccine and mAb randomized, controlled prevention efficacy trials, healthy HIV-1 negative volunteers are enrolled and monitored for the primary endpoint (HIV-1 infection). The first five HIV-1 VE trials used 6-monthly HIV-1 testing [19,20,21,22,23], whereas the most recently completed VE trial (HVTN 505) used three-monthly testing [24]. Three-monthly testing is also being used in the ongoing HVTN 702 VE trial (NCT02968849) [25] and the ongoing HVTN 705 VE trial (NCT03060629) [26]. Two other ongoing efficacy studies conduct HIV-1 testing on a monthly (four-weekly) basis: the antibody mediated prevention (AMP) studies of VRC01 at two dose levels vs. placebo (protocols HVTN 704/HPTN 085 (NCT02716675) and HVTN 703/HPTN 081 (NCT02568215) [27]. These are the first efficacy studies of a monoclonal antibody for prevention of HIV-1 infection.

Improving infection time estimation for HIV-1 prevention efficacy trials is important for a variety of secondary objectives of these trials:
(1)Time-dependent marker correlates of risk (CoR) of HIV-1 infection: For studying the correlates of HIV-1 risk, a case-cohort or case-control study design can be used to measure a time-varying potential correlate (marker) of interest as near as possible prior to the time of HIV-1 acquisition for all HIV-1 infected cases. Moreover, for a random sample of participants who complete follow-up testing as HIV-1-negative, the marker(s) is measured at all longitudinal sample time points (e.g., this design was employed in the VaxGen HIV-1 VE trial [20,28] and the Partners in PrEP prevention efficacy trial [29] and is planned for the AMP prevention efficacy trials [27] as well as for the HVTN 702 and HVTN 705 VE trials). In AMP one key marker of interest is VRC01 serum concentration measured by ELISA or serum neutralization titer against a standard panel of viruses by a neutralization assay; population pharmacokinetics/pharmacodynamics (PK/PD) models can be used to provide low-error unbiased estimates for the VRC01 concentration in infected individuals [30], given an accurate estimate of the date of infection. An important goal of the AMP trials is to characterize the relationship between a person’s VRC01 concentration and their instantaneous risk of HIV-1 infection. Identification of a serum neutralization threshold associated with (very) low risk of HIV-1 infection would provide valuable guidance for future vaccine development. What makes it challenging to pinpoint a marker’s value at infection is uncertainty in the date of infection. Even with monthly HIV-1 testing with high adherence to the testing schedule, the estimation methodologies that we previously employed for evaluation of HIV-1 VE trials are inadequate for the requirements of the AMP studies. In Appendix A we illustrate the amount of increase in statistical power to detect such a CoR in the AMP studies that we expect to result from reducing the error in the infection time estimator (Appendix A) using our previously applied approach [31].(2)“Sieve analysis”: How the level of vaccine/prevention efficacy depends on genotypic characteristics of HIV-1 at the time of acquisition: Sieve analysis provides another tool to detect and evaluate correlates of vaccine protection, based on the comparison of viruses that infect placebo recipients with the viruses that infect vaccine recipients, despite the protective barrier induced by vaccination [32]. An ongoing challenge for sieve analysis is that the determination of HIV-1 genetics at the time of HIV-1 acquisition is of fundamental importance for discriminating true sieve effects from post-acquisition effects. That is, whether observed viral genetic differences (across treatment groups, vaccine vs. placebo) can be interpreted as differential blockage of acquisition of incoming variants (a true “sieve effect”) vs. as resulting from differential evolution post-infection of similar starting viruses, resulting for example from effects in which vaccine-induced anamnestic responses impact the early evolution of HIV-1 prior to diagnosis (and sampling for sequencing) [33]. This issue has been critically important in the interpretation of sieve effects for all HIV-1 sieve reports to date [34,35,36,37,38]. Statistical methods have been developed that require the ability to determine which HIV-1 infection events are diagnosed very early prior to significant post-infection evolution [39,40,41]; additional research is needed to ensure that the methods optimally incorporate state-of-the-art infection time estimators.

Sieve analysis also requires knowledge of the multiplicity of infection—i.e., the number of founder viruses that establish a given infection—because all founding lineages that infect the vaccine vs. placebo recipients must be identified in order to comprehensively identify the genetic differences between the viruses from the two groups. To develop and evaluate methods that estimate the time of HIV-1 infection and founder multiplicity, we required a “gold standard” dataset of longitudinal HIV-1 sequence samples including (1) a very early sample so that the closest to “true time” since acquisition and founder multiplicity could be determined, and (2) later samples available in a time frame relevant for application to clinical trials (1–6 months post-infection). Such a dataset has not previously been available; however, we recently obtained genome sequences satisfying these criteria from samples collected in the RV217 [42] and CAPRISA 002 [7] trials. As shown in Table 1, in every case these included multiple time points per person: one time point with sequences from samples that were collected within 14 days of estimated infection, and sequences subsequently collected from approximately 1–2 and six months after infection.

Here we report the results of comparing predicted infection time values and multiplicities for each participant to gold-standard values that were established prior to conducting this analysis using separate data (based on sequences sampled earlier, see Methods). We compared analysis methods that fit Poisson models to measured mutation counts, in addition to methods that fit phylogenetic trees to account for correlation that arises through shared ancestry. We call results from these methods “uncalibrated estimates” because we employed the corresponding software using default parameters. We then sought to find linear model fits using gold-standard times (“calibrated estimates”) that improve these results to optimize prediction of gold-standard times in held-out samples. We found that the accuracy of both infection time and founder multiplicity prediction was improved by using linear regression modeling to shift and scale the uncalibrated estimates from individual prediction tools.

## 2. Materials and Methods

### 2.1. Studies, Participants, Diagnostic Testing and HIV-1 Sequencing

The RV217 Early Capture HIV Cohort Study (ECHO) and the CAPRISA 002 study were prospective studies of acute HIV-1 infection (Table 1). RV217 was conducted in Kenya, Tanzania, Thailand, and Uganda and enrolled HIV-negative participants at high risk of HIV-1 infection; data from 36 (17 Thai men and 19 Ugandan and Kenyan women) were available for our analysis [42]. Blood samples were obtained by fingerstick twice-weekly from participants who met at least one high-risk criterion; larger volume blood samples were collected every 6 months. Blood samples were obtained at approximately one week, one month and six months after testing positive from an APTIMA HIV-1 RNA test. Near full-length HIV-1 sequences were obtained from plasma samples from infected participants by single genome amplification and sequencing [43], averaging 10 near full-length genomes per participant (Table 1). CAPRISA 002 was conducted in South Africa and enrolled 245 uninfected high-risk women who predominantly self-identified as sex workers [44]. Blood samples were obtained from 21 women at approximately one week, two months, and six months after testing positive from at least two different HIV antibody tests. HIV-1 sequences of the Env V3 region were obtained by Illumina deep sequencing for each participant enrolled in the study. Written informed consent was obtained from all participants in both RV217 and CAPRISA 002 (see the Ethics statement in the Appendix A).

### 2.2. Sequence Data Pre-Processing, Hypermutation Detection, and Recombination Detection

We employed two pre-processing steps that were implemented as part of the software pipeline hiv-founder-id (http://github.com/pedlefsen/hiv-founder-id) described below, which excluded hypermutated and recombined sequences. For hypermutation detection, we used our reimplementation of the software tool Hypermut 2.0 [45,46] to identify signatures of APOBEC3G-mediated hypermutations and removed sequences with sufficient evidence of these signatures. The software is available at https://github.com/pedlefsen/hiv-founder-id/blob/master/removeHypermutatedSequences.R. We used a p-value threshold of 0.1 (for consistency with the threshold used in the previous study of CAPRISA 002 acute sequences [47]). For recombination detection, we used the online RAPR tool (http://www.hiv.lanl.gov/content/sequence/RAP2017/rap.html) [48] to identify sequences that were recombinants of other sequences in the sample and excluded the recombinant sequences from downstream analysis.

### 2.3. Infection Time

We define the time since infection, *t*, as the number of days between the infecting exposure event and sample collection for sequencing and viral load measurements. For these data sets, *t* is never known exactly due to the eclipse phase (between exposure and detectability) [49] and patient-specific diagnostic marker variability. Additional uncertainty is introduced because HIV-1 testing was not conducted daily. Even with the very frequent sampling conducted in the RV217 cohort study, we can at best identify the closest to true value of the date of first RNA-positivity to within the 2–3 days (outside of the eclipse phase) based on the last RNA-negative and first RNA-positive sample. gold-standard infection dates were determined by applying a standard algorithm based on the date of first positive diagnosis, date of last negative diagnosis, and the diagnostic test applied [1,3,50,51,52]. Briefly, the infection date is computed as the midpoint between these dates, so we refer to this method for estimating the infection date as the “center of bounds” (COB) method, although in the case of these RV217 sequences the procedure was more accurately calculated than previously available due to very tight sampling. For the CAPRISA 002 data, these dates correspond to those published previously [47], which also employed the COB method. Comparisons of estimated dates of infection (using data from later timepoints) to these gold-standard estimated dates were made by counting the days separating the two estimated dates.

We computed the bias, which is the average difference across individuals between the predicted and gold-standard values, and the cross validated root mean squared error (RMSE), which is the square root of the average of the squared differences, both before and after the calibration process described below. If any estimated time falls outside of the bounds on the infection date, the nearest bound is used instead.

### 2.4. True and Artificial Diagnostic Bounds on the Date of Infection

The bounds on the infection date are to be determined from diagnostic test results in the setting of the envisioned application, but for the purposes of our study we also created more realistic artificial bounds. In order to maximize estimation precision, our study required sequence data that were collected very early from acutely encountered infections that were then followed and sequenced longitudinally—we thus know the infection dates to a higher degree of precision than is typical. In this setting the first positive diagnostic test result and last negative visits are not relevant to what we expect in the envisioned application settings. Instead a longer sampling interval is better reflective therefore we use a later longitudinal time point (~1–2 month between diagnoses is relevant for AMP studies, around six months is relevant for historical HVTN studies). We forego use of the true diagnostic intervals, which would lead to unrealistic expectations. Instead, we choose a larger artificial diagnostic bound interval to match the longitudinal datasets similar to clinical trial intervals. For the ~1–2-month data we selected interval widths (between last negative and first positive diagnosis date) from true distribution of MTN 003 intervals, and we selected the ~6-month interval widths from the HVTN 502 intervals. For each participant at each of the time ranges, we selected artificial interval widths uniformly at random with replacement from the corresponding trial data, and we determined start dates of each artificial interval randomly, such that the there is a uniform distribution for the location of the gold standard date of infection within the artificial diagnostic bounds interval.

### 2.5. PFitter Estimate of Days Since Infection

The Poisson Fitter (PFitter) software employs a maximum likelihood technique to fit parameters of a Poisson process representing a molecular clock star-like phylogeny model to an aligned set of sequences from a single patient [53]. The target parameter of this calculation is the time elapsed between sampling time and infection, *t*. Other parameters can be summarized as the mutation rate per day, which in PFitter is 1.19e^−5^ unless otherwise specified (and we use this default value). This is computed from the mutation rate per generation (ε = 2.16 × 10^−5^) [54], the basic reproductive ratio (R_0_ = 6) [55], and generation time (τ = 2 days) [56]. With these parameters, PFitter estimates the time since infection *t* using the formula $\hat{t}$ = (0.9065 * $\hat{\lambda }$/εn) − 0.205 [57], where *λ* is naturally estimated by $\hat{\lambda }$, the maximum likelihood rate of a Poisson fit to the number of mutations k observed out of n total residues in the input alignment: k~Poisson(*λ*). However, the actual fit is based on k’ ~ Poisson (2$\hat{\lambda }$), where k’ is the total Hamming distance across all pairs of input sequences, since under the star-like model these models share the same value of λ. Note that since the resulting value of $\hat{t}$ is rounded to the nearest integer, the constant term −0.205 is effectively negligible. Thus, PFitter’s estimate of t is approximately equal to c × ($\hat{\lambda }$/n), where c = 0.9065/ε represents the effective mutation rate per base per day. Thus multiplying $\hat{t}$ by any constant x is nearly equivalent to calling PFitter with an alternative epsilon value ε’ = ε/x, a fact that we utilize during the calibration process to refit mutation rate values without recomputing $\hat{t}$, as discussed below. Instead we need only compute $\hat{t}$′ = $\hat{t}$ × x. During the calibration of the simplest scale-only models (Appendix A), we find optimal values of x to fit the data, and thus optimal mutation rate parameters.

### 2.6. Variations on the PFitter Estimator $\hat{t}$ of t: (syn) and (w/in clusts)

PFitter works well when the model fits, meaning there is both neutral evolution (drift) and independence of mutations across sequences (star-like rather than branching topology of the phylogenetic tree). This model does not account for multiple founders, superinfection, compartmentalization leading to identity by descent, or selection. We employed two simple modifications of the Poisson Fitter software alone and together to address small violations of these assumptions. The Synonymous PFitter (“PFitter (syn)”) approach simply pre-processes inputs before calling PFitter, first masking out (replacing by gaps) codons that translate to different (non-synonymous) amino acids than the estimated ancestral codon (the consensus of the alignment). The Within-Clusters PFitter (“PFitter (w/in c)”) approach first applies a simple clustering procedure to the sequences (described below), then calls PFitter with a modified pairwise distance matrix that omits distance values whenever the two sequences are in a different cluster. This required a minor modification to the calculation of the estimator to accommodate missing intersequence distance values. The version of PFitter with these modifications has been contributed to the maintainers of PFitter for consideration of this modification and is available at https://github.com/pedlefsen/hiv-founder-id/blob/master/PFitter.R. We also employed a combined version (“PFitter (w/in c + syn)”) which first clusters the sequences, computes per-cluster consensus sequences and uses these to mask non-synonymous codons, then proceeds to call PFitter with the between-cluster pairwise Hamming distances (HDs) missing.

### 2.7. Clustering Sequences for the Within-Clusters PFitter Method

To cluster sequences, we employed a simple strategy that was independent of the determination of founder multiplicity: first we calculated the pairwise Hamming distances across all sequence pairs using the sub-alignment that contains only informative sites (these are columns of the alignment at which at least two sequences contain the same non-consensus mutation). We then employed a simple agglomerative hierarchical clustering procedure (using the hclust method in R using the “average” agglomeration method, so this is UPGMA clustering [58]) and then trim the resulting tree into one or more clusters by applying the cutreeDynamic method from the dynamicTreeCut package [59] with the “hybrid” cutting method, no limit on tree height, and a minimum cluster size of 1. These are accomplished using the perl and R code available at https://github.com/pedlefsen/hiv-founder-id/blob/master/clusterInformativeSites.pl and clusterInformativeSites.R.

### 2.8. PrankenBeast

PrankenBeast (also known as PREAST) is a software pipeline for estimating founder sequences and infection times using a combination of the software Prank [55] and Bayesian Evolution Analysis by Sampling Trees (BEAST) [56]. First, a multiple sequence alignment is made using Prank. To estimate dates when multiple founders may be present, the phylogenetic tree inferred by Prank is split at the root and the leaves on each half are used to infer separate founder trees in BEAST. Time of infection is inferred with BEAST using a strict molecular clock and constant population size as priors, using tree height as an estimator of *t*. Tree height is constrained to be consistent with sample dates on the sequences. PrankenBeast is available at https://github.com/matsengrp/PREAST.

### 2.9. Founder Multiplicity Characterization

We focused on the dichotomy of single- vs multiple-founder infection due to insufficient gold-standard information for use in estimating specific infection multiplicity counts above 1. Dichotomous founder multiplicities were determined from these and additional sequence data as well as (in RV217) non-sequence data in collaboration with the study virologists—in the case of CAPRISA 002, these agreed with previously published results based on SGA sequences obtained from the same early timepoint samples [47]. Estimated multiplicities were compared to these gold-standard multiplicities by computing the area under the receiver operating characteristic (ROC) curve (AUC). The ROC curve is a plot of true positive rate against false positive rate; in this case the curves are trivial because the estimators are binary. We computed the AUC using the R package ROCR [60]. For a single binary predictor, the AUC of the ROC is equivalent to 0.5 + 0.5 * abs(P_1_−P_0_), where P_1_ and P_0_ are the fractions of cases predicted to be multiple-founder infections among those truly multiple-founder and among those truly single-founder, respectively. Its (effective) minimum is 0.5, and it is maximized (at 1) when the sensitivity and the specificity are both 1.

### 2.10. Rolland HVTN Method for Determining Founder Multiplicity

To estimate the indicator of a multi-founder infection, we used a method that is typically employed in HVTN trials and elsewhere, which was developed by Morgane Rolland [38]. Using the InSites software from DiveIn [61], we computed the ratio of informative sites [62] to private sites (those with non-informative mutations), and also the sequence diversity (mean pairwise Hamming distance). The infection is estimated as multi-founder if both the ratio and the diversity exceed a pre-determined threshold. We tuned these thresholds on the ~1-week timepoint sequences to achieve a perfect classification of the gold-standard multiplicities: for the RV217 NFLG sequences we used a ratio threshold of 0.85, and for the CAPRISA 002 V3 sequences we used a ratio threshold of 0.33. In both cases we used a diversity threshold of 0.001, the value employed for analyzing HIV-1 sequence data from breakthrough infections in HVTN 505 [38].

### 2.11. Tests for Star-Like Phylogeny or Founder Multiplicity

We also included two tests of a star-like phylogeny based on a single time point post-infection, provided by PFitter [53], and a novel method, DS StarPhy Test. PFitter performs one formal and one informal hypothesis test to detect violations of the assumption of a star-like phylogeny. The formal test (fits) evaluates whether the data are consistent with the star-like phylogeny model using a Chi-squared test to compare the expected to the observed distribution of inter-sequence Hamming distances. The expected values are computed under a Poisson model with a rate fixed at twice λ_c_, where λ_c_ is the estimate of the rate of the Poisson process of mutation events that is calculated from the distances between each sequence and the consensus sequence. In the informal “convolution test” (*is star-like*), the data are declared to “not follow a star-like phylogeny” when there is more than 10% cumulative difference between the mass distribution of the inter-sequence HD histogram and the expected inter-sequence HD distribution. PFitter calculates the expected number of inter-sequence pairs having each possible HD value by convolving the observed HD distribution of distances to the consensus sequence [53,57]. This procedure, while not itself a formal statistical test, defines a sensible strategy for evaluating the hypothesis of a star-like phylogeny by asking whether the inter-sequence HDs are consistent with a convolution of the consensus HDs. Note that these are not designed as tests of multiple-founder infections, as there are other reasons why the star-like phylogeny model might be a poor fit (such as if the data follow a branching phylogeny model instead, as would be the case in later infection). The DS StarPhy Test employs Dempster-Shafer Analysis [63], a fiducial methodology that generalizes Bayesian inference to cases lacking prior distributions, to implement a simple variant of Pfitter’s fits test, which evaluates the assertion that under a star-like phylogeny, the inter-sequence Hamming distance rate is twice the distance-to-consensus rate λ_c_. Details and an implementation of this method are available in the hiv-founder-id github repository (DSStarPhy.Rnw).

### 2.12. Statistical Methods for Calibrating Predictors of the Indicator of a Multiple-Founder Infection

All of the above-mentioned methods for multiple-founder identification are considered “uncalibrated” because we have not attempted to optimize their parameters. We employed a leave-one-out cross-validation procedure [64,65] to fit regularized logistic regression models (LASSO) [66] to data sets with one person’s values held out at a time, and iterated over all people’s data to obtain a set of binary predictions of the Rolland-method indicator of multiple-founder infection based on the best-fitting models. In these models we always included an intercept, and because these values will be available in the anticipated application setting of HIV-1 prevention efficacy trials, we explored the value of including the log_10_ plasma viral load (lPVL) and the days elapsed between sample collection and infection diagnosis, and interactions between these variables and each binary uncalibrated predictor (Rolland HVTN, PFitter fits, etc.). We also allowed the inclusion of several additional predictors (Appendix A), from which the LASSO procedure was allowed to sub select.

The choice of leave-one-out (versus k-fold) cross-validation was made because of the small sample sizes of these datasets (Table 1; *n* = 18 to 36). Within each leave-one-out iteration we employed the cv.glmnet procedure in R’s glmnet package with arguments grouped = FALSE and nfold = *m*, where *m* = *n*–1 is the size of the leave-one-out data set. These options cause the selection of the LASSO tuning parameter at each iteration to employ the same procedure at one additional (inner) level of recursion. The tuning parameter is selected by optimizing prediction of the held-out value in another leave-one-out procedure iterated over the *m* values not held-out in the outer iteration.

### 2.13. Statistical Methods for Calibrating Predictors of Infection Timing

Similar to the methods for identifying multiple-founder infections, all the above-mentioned methods for estimating infection time are considered “uncalibrated”. As noted above, the Poisson Fitter estimate depends on a mutation rate parameter through a simple linear relationship with a negligible intercept term and a slope coefficient that can serve as a tuning parameter for scaling the mutation rate. To determine if using a different mutation rate would reduce the prediction error, we fit linear models regressing the gold-standard infection times on the estimated days since infection. We accomplished this using leave-one-out cross-validation [64,65]: using the gold-standard infection times we fit parameters of regression models to all of the data except for those from one person, and used the fitted model to estimate the infection time for that held-out person, then iterated this procedure over all people to obtain a set of predicted values. We explored using regularized regression (LASSO) [66] to search larger models that include the covariates discussed above for calibrating estimators of multiplicity.

### 2.14. Software Pipeline

Briefly, the above methods have been implemented as an automated pipeline with the goals of maximizing the reliability and reproducibility of HIV infection timing and multiplicity estimates, while supporting audit and follow-up review by bioinformatics experts (Further details available: Appendix A). The source code for the infection time and founder multiplicity estimation pipeline is available at https://github.com/pedlefsen/hiv-founder-id, and the pipeline itself has been packaged into a docker container available at https://hub.docker.com/r/philliplab/hiv-founder-id with a working example available in the container that is accessed by running the script /home/docker/example_docker.sh.

## 3. Results

### 3.1. RMSE and Bias of Center-of-Bounds (COB) Estimates of Infection Time

We set out to determine the bias and RMSE associated with COB (HVTN-standard) estimates of infection time. As described above, our study necessarily employed artificially shifted-apart diagnosis interval endpoint dates, with interval lengths sampled directly from times between last negative and first positive diagnosis dates among infected participants in clinical trials of HIV prevention efficacy. The estimates of bias for this procedure are impacted by our unbiased random placement of these intervals, and the RMSE values largely reflect the sampled distribution of the interval lengths. Note that these interval lengths have distributions that are observed in actual clinical trials, and as such they include some right skew and high outliers, reflecting that some infected participants are missing one or more of their scheduled visits for diagnostic testing (Table 1).

Appendix A shows that the COB-estimated times are inaccurate and imprecise. Overall, the COB-estimated infection times correlate poorly with the gold-standard infection times within each study, with RMSE 24.7 and 41.1 days for the COB estimates from the 1‒2-month and 6-month category, respectively for RV217, and with RMSE 19.6 and 41.5 days for CAPRISA 002. However, across studies the COB is informative for determining whether a given sequence is from the 1‒2-month or the six-month time category. The CAPRISA 002 V3 samples in the 1‒2-month category were sampled later than the corresponding NFLG sequences from RV217 in the 1‒2-month category (Table 1, Appendix A), although the predicted times are largely overlapping across these data sets. For the 6-month category, the infection times have approximately the same distribution across these two data sources. For building prediction models as described below, pooling data sources increases the robustness of the model; we therefore pooled the data across the two regions/data sources for the six-month time category. However, for the 1‒2-month category, the later gold-standard infection times of the CAPRISA 002 data introduced a confounding factor for modeling the pooled data.

### 3.2. Prediction Error of Sequence-Based Estimators of Time Since HIV-1 Infection is Improved with Calibration

We next compared the prediction error of several estimators of infection time, including the HVTN-standard COB estimator, estimators that employ HIV-1 sequence data to fit a molecular clock [PrankenBeast and Poisson Fitter (PFitter)], and variations of the PFitter estimator [Figure 1a]. The PrankenBeast estimator is an application of the Bayesian phylogenetic tree modeling software package BEAST that accounts for alignment uncertainty using the statistical alignment software package Prank. The PFitter estimator is simpler as it fixes the alignment and the tree structure so that its estimate relies solely on the expected daily increase of virus genetic diversity in the acute stage of infection (i.e., prior to the onset of the host immune response). In addition to using PFitter with its default values, three variations were used: the “syn” PFitter method is applied to the nucleotide alignment after first masking out codons that result in non-synonymous amino acid changes from the consensus codon; the “w/in c” PFitter method first clusters the data and uses this to modify the distance matrix input to Poisson Fitter such that it only includes entries between pairs of sequences assigned to the same cluster (distances between sequences from different clusters do not enter the computation). We also included a combined version “PFitter (w/in c+syn)” which first clusters the sequences and then uses the within-cluster consensus sequences to determine which codons are non-synonymous for masking. These latter changes were motivated by the observation that exceptions to a star-like phylogeny and molecular clock model can arise due to multiple-founder infections or early compartmentalization (leading to multiple approximately independent star-like phylogenies), and to selection operating on a locus (which typically operates on the amino acid so can be detected by non-synonymous codons).

Overall, as expected, all six of the estimators showed improved performance for the 1-month samples compared to the six-month samples (Figure 1a–d). The estimators showed similar performance when using NFLG sequences versus V3 sequences, except that the PrankenBeast estimates were much better for the NFLG sequences than for the V3 sequences (Figure 1a,c vs. Figure 1b,d). Uncalibrated estimators performed poorly. Regarding the performance of the different estimator methods, due to study design limitations we are unable to formally statistically evaluate comparisons of our sample-based estimates of the bias and RMSE (nor of ROC-AUC), thus we limit our interpretation to observations about the results of prediction from our limited samples. We observed that the COB estimator had an RMSE of about 20 days for 1‒2-month and 40 days for six-month sequences (Figure 1b,d), despite an upward bias (toward estimating the infection as older than it is) in both cases. Additionally, for the 1‒2-month NFLG samples, the COB estimates were occasionally relatively far from the closest to true value, as illustrated by the outlier points above the whisker. The PrankenBeast and PFitter estimators showed similar performance to the COB estimator for the 1‒2-month NFLG samples, but generally exhibited relatively strong bias in the opposite direction (i.e., tended to estimate that the infection had occurred later than it had actually occurred) for the six-month NFLG, 1‒2-month V3, and 6-month V3 samples (Figure 1b,d). The three variations of the PFitter method did not reduce RMSE appreciably or consistently. The greater bias of the modified versions of PFitter compared to the original may suggest that the default mutation rate in PFitter is not well-calibrated for application to these modified inputs, since for instance the effective mutation rate at the subset of only synonymous positions may differ from the effective overall rate, since these rates result from a mutation-selection process with varying selection effects across positions.

The PFitter estimate of days since infection factors into a data-estimated Poisson intensity multiplied by a scalar representing the effective mutation rate. As the default mutation rate was derived from first principles and parameterized by separately-estimated factors (see Methods), we sought to determine if using a different mutation rate would reduce the prediction error. To explore the extent to which underperformance of these timing estimators is due to mutation rate misspecification, we fit a model with a single parameter to scale the PFitter result, with no intercept (Appendix A). We found that for PFitter, the estimates were improved by scaling PFitter’s estimated days since infection downward, while the results of running PFitter after first masking codons exhibiting non-synonymous variation support that the combined effects of mutation and selection impact the effective mutation rate. PrankenBeast results also showed improvement after this type of calibration. Since other covariates could potentially improve estimates further, we endeavored to improve the predictive capacity of the estimators by calibrating the mutation rate using person-specific models that include lPVL and diagnosis date (the lower bound on the time since the infection). The calibration procedure is described in detail in the Methods section. The performance of the calibrated estimators is shown in Figure 1e,h.

Strikingly, calibration considerably improved all estimators, reducing bias to near zero and reducing root mean squared error (RMSE) to 5–10 days. This improvement due to calibration was consistent across 1–2-month and ~6-month samples, NFLG and V3 samples, and estimators. Linear models regressing the gold-standard infection times on the estimated days since infection were used to determine whether use of a different mutation rate would reduce the prediction error. This was performed using leave-one-out cross-validation [64,65]: using the gold-standard infection times we fit parameters of regression models to all of the data except for those from one person, and used the fitted model to estimate the infection time for that held-out person, then iterated this procedure over all people to obtain a set of predicted values. K-fold cross-validation was not possible for these data due to small sample sizes (see Methods), so while these predictions are expected to be relatively unbiased, performance of the estimator may vary widely when applied to other datasets; see the Discussion below. We found that the performance of larger models evaluated using LASSO did not compete with the simple models that we manually explored, which included only the log_10_ plasma viral load, the time since diagnosis, and interactions between these parameters and the days since infection output from PFitter or PrankenBeast. For the sequences from the earlier time-point category (1–2 months post-infection). Figure 1 shows results for the CAPRISA 002 sequences that were estimated separately from those of the RV217 sequences; for the ~6-month time category we present results from a combined model that we trained with both datasets, lending additional robustness to the interpretation of the results. We conclude that, with calibration, simple estimators of time of HIV-1 infection, such as the unmodified Poisson Fitter estimator or the COB estimator, perform as well (or nearly so) as more sophisticated methods.

### 3.3. Multiplicity Assessment is Improved by Calibration with LASSO

All analyses so far were restricted to assessing the performance of various estimators of time of HIV-1 infection. However, about 20% of sexually-transmitted HIV infections are established by multiple founder lineages [47,67,68,69]. Since infection with multiple founder lineages can limit the accuracy of identifying recent HIV infections [9] and negatively affect timing estimates, the identification of multiplicity is a crucial first step in the standardization of timing estimates. We assessed the performance of seven different estimators of founder multiplicity in terms of correctly classifying the founder virus(es) as single or multiple as described in the methods section.

To assess the performance of each estimator of founder multiplicity, we computed the area under the receiver operating characteristic curve (AUC) of the two-by-two contingency tables of gold-standard versus predicted indicators of a multiple-founder infection. The results are shown in the orange bars in Figure 2. Similar to the timing estimators and as predicted, performance of all 7 estimators was generally better using the one-month sequences compared to the six-month sequences. In addition, the performance of all seven estimators appeared to be comparable between NFLG vs V3 sequences. With respect to the individual estimators, PFitter’s *is star-like* indicator outperformed its *fits* estimator for the 1–2-month data, and the modifications (w/in clusts) and (syn) generally improved the prediction accuracy of these indicators. We found that without calibration, the PFitter estimates performed similarly to the HVTN standard estimator, but in each case (except for the V3 1M sequences) at least one of the modified PFitter estimators matched or exceeded the performance of the HVTN estimator.

Since calibration yielded such striking improvement in the performance of the HIV-1 time of infection estimators (Figure 1), we next explored calibrating the estimators of founder multiplicity. For each estimator, the AUC was calculated on the contingency table of predictions made on held-out data during LOOCV, where for this analysis we pooled all data across times and regions (and held out one person’s data, for both time points, per iteration) as described in Methods. We employed logistic regression models to fit the indicator of multiple founder infection to each of the binary predictors from Rolland HVTN and PFitter and to other covariates using the LASSO variable selection procedure applied within each cross-validation iteration, selecting from a suite of additional predictors (Appendix A) while always including an intercept, lPVL, and diagnosis time. Final selected predictors and their coefficients are shown in Appendix A.

The performance of the calibrated models is shown in Figure 2 (turquoise bars). Similar to the timing estimators, calibration yielded strong improvements with highly accurate predictions (ROC-AUC > 0.85) in most cases. Calibration eliminated method-to-method performance variation for the assessment of the V3 sequences (Figure 2b,d) and mostly eliminated this variation for assessment of the NFLG sequences (Figure 2a,c), providing initial support for employment of a calibrated predictor for the analysis of founder multiplicity in HIV-1 prevention efficacy trials, using a logistic regression model with lPVL, diagnosis time, and additional per-sample statistics computed by the various software tools that are incorporated into the pipeline. We found that inclusion of (lPVL) and diagnosis date were important for classification accuracy so the results we report are based on models that include these predictors. The contributions of other predictors were far greater; however, the most informative input to the multivariate calibration model was the pairwise sequence diversity, which inflates when there are multiple founders. Interestingly, the output of DS StarPhy Test contributed to the model through the difference between evidence of a fit when applying the method to unaltered inputs (“DS StarPhy R”) versus to the data after masking nonsynonymous codons and then clustering the sequences (“(w/in clusts) (syn) DS StarPhy R”). This is sensible, because in the case of a multi-founder infection, clustering the sequences first and computing only within-cluster fits should improve the estimate.

### 3.4. Calibration, Considerations and Results Summary

A major improvement in accuracy resulted from the calibration process; as stated we provide a version of the developed pipeline that outputs the estimates of days since infection and of founder multiplicity using calibrated estimators based on the model fits. Briefly, the improved infection timing and multiplicity estimation employed maximum likelihood estimators that shift and scale (calibrate) estimates by fitting “true” infection times (gold standard) and founder multiplicities to a linear regression model. The tool optionally accepts viral load inputs, which we expect to have available in the anticipated application to the analysis of infections occurring during HIV-1 prevention efficacy trials. If these data are absent, the tool employs alternative calibrations using models built without (lPVL) data. Likewise, if known, the user can provide a time category and/or a region designation (NFLG or V3) to use calibration models that were optimized for those settings. Results include an output table containing statistics encompassing estimates of founder multiplicity, nucleotide and amino acid founder (ancestral) sequences (not discussed here, these are simply the consensus sequences of the clusters), and time since infection. The tool also provides estimates of the standard errors of the outputs based on the cross-validation error [70].

Using Poisson models of measured mutation counts and phylogenetic trees, we analyzed longitudinal HIV-1 sequence data together with diagnostic and viral load data from the RV217 and CAPRISA 002 acute HIV-1 infection cohort studies. We used leave-one-out cross validation to evaluate the prediction error of these calibrated estimators versus that of existing estimators. Calibration considerably improved all estimators of time since HIV-1 infection, reducing bias to near zero and reducing root mean squared error (RMSE) to 5–10 days. Calibration of multiplicity assessments yielded strong improvements with highly accurate (ROC-AUC > 0.85) predictions in most cases.

Despite our application of cross-validation to yield representative prediction errors, there is likely still a high degree of overfitting to these data, and validation is a necessary next step. The structure of these data constrained our ability to employ alternative forms of cross-validation, especially relatively small sample sizes within each time/region dataset (Table 1). A particularly subtle but important point is that the calibrated models for infection time presented here include intercept terms in the linear regression model for predicting days since infection, and therefore these models lose interpretability as a simple rescaling of the mutation rate. Instead, these models represent a hybrid between (a) linear models for predicting infection time simply from (lPVL) alone (which can explain most of the variability, with R^2^ values above 0.80) and (b) a mutation rate model which applies a scalar multiple to the estimate that results from the center of bounds approach, or from methods that use sequence data (Poisson Fitter, BEAST, etc.). The presented models in Figure 1 yielded the lowest RMSEs despite having low R^2^ values (less than 0.20) when evaluating the fit on all of the data, and wide confidence intervals around the estimators (the predictive power of these models relies mostly on the intercept). Models with much higher R^2^ values (above 0.98) can be obtained by omitting the intercept (Appendix A). These models yield interpretable coefficient estimates while retaining the low bias of the best models calibrated with intercepts but have higher RSMEs.

Our ultimate goal is to select parameters that will minimize error of predictions in specific settings, such as the AMP studies. It is worth noting that the training sets differ in many respects (Table 1). Our CAPRISA 002 subset is composed of Southern African young women with clade C infections sequenced using short-read (V3) targeted amplicon deep sequencing technology, whereas our RV217 subset includes men in Thailand and women in East Africa with a mix of infecting HIV-1 subtypes, sequenced using endpoint dilution Sanger single genome amplification at a constant shallow depth (~10 nearly full length genome equivalents). The consistency of estimation improvement through calibration across these very different cohorts lends support that the approach to calibration will also prove valuable in future settings. However, further validation is needed to evaluate the robustness of these specific calibration model fits when applied to different sequencing technologies, HIV-1 subtypes, and sampling times. We recommend further validation with data more similar to the target setting, and potentially further calibration, before employing these calibrated models in practice.

## 4. Discussion

Our study addresses an important and globally relevant need to improve assessment of HIV prevention and treatment intervention through improved approaches to estimate timing of HIV infection and founder multiplicity. The unique granularity of the viral sequence data generated in recent HIV-1 acute infection trials has provided a valuable opportunity to build and evaluate our comprehensive pipeline, which brings together available tools [71,72] and provides statistical calibration for estimators of the time of HIV-1 acquisition and founder multiplicity yielding enhanced accuracy and in a systematic and reproducible way. The data available for this study were limited to natural infection; future work is needed to evaluate how infection timing estimators perform in the presence of any intervention. To enable application of the calibrated estimators developed here for correlates of risk and sieve analyses in the AMP trials, it will be necessary to either effectively bridge data from external gold-standard studies such as RV217 and CAPRISA to AMP, or to ascertain approximate true infection dates for an identified subset of HIV-1 infected participants within AMP. The AMP trials apply three types of HIV-1 diagnostic kits to participant samples—HIV-1 Antibody tests, HIV-1 RNA tests, and HIV-1 RNA ultrasensitive tests – and one approach to defining this gold-standard subset would select infected participants whose first HIV-1 positive sample has diagnostic testing results HIV-1 Antibody Negative, HIV-1 RNA Negative, and HIV-1 RNA Low Copy Positive. Further research is needed for optimizing within-trial calibration of infection timing estimators. Another challenge posed to application of our methods to the AMP trials is that the presence of VRC01 antibody in some participants of these studies will have unknown effects on the diversification rate and on the diagnostic inputs. Future work is needed to evaluate how infection timing estimators perform in the presence of different interventions, acknowledging that data do not currently exist to effectively evaluate and calibrate in the presence of VRC01. Another potential application of our estimators evaluates timing of HIV-1 infection in individuals infected while taking other prophylactic interventions such as PrEP, which has been reported to affect laboratory markers/diagnostics of HIV-1 infection [73,74]. The delayed seroconversion in individuals on PrEP has been proposed to be due to lower viral load [73] which impacts the diversification rate, raising the question as to whether our estimators can be effectively calibrated to the lowered diversification rate of PrEP users and maintain performance. The answer to this question remains to be studied experimentally.

Mathematical modeling allows us to simulate the gain in unbiasedness and precision/power through using a more precise infection time estimator when addressing important questions relevant to trial endpoints of ongoing vaccine and passive immunoprophylaxis efficacy studies. Our results illustrate the need for improved methods when idealized sampling for diagnostic testing is not possible in real-life settings. The power gain, cost savings, and improved validity that is achieved when the estimated infection time is known to within 5–10 days of the true infection time supports that this is a research area that needs to be prioritized towards the goal of identifying an immune correlate of protection for an effective HIV vaccine.

Excellent work has been done by other groups regarding the development of estimators of HIV-1 infection timing [14,17,75] for acute infection, classification of infections as acute/chronic, and for chronic infections. The focus of our study is day-of-infection estimation for the acute incident infections seen in HIV-1 prevention trials. A major advantage of our approach is that we have employed a relatively large and relatively diverse set of HIV-1 acute infected participant data with known infection times and longitudinal sequencing, which we have used for evaluating and for calibrating the infection timing and founder multiplicity estimators. Our study is limited by the data that were available, and the calibrated models where the best prediction performance was seen may not be as robust as those with lower performance. The two datasets we analyzed were not large enough to facilitate fully honest estimates of prediction error generalizing to new samples; therefore, when additional acute infection datasets become available it will be important to conduct such analyses. Appendix A shows results of applying models that exclude the intercept, which we expect to be more robust. Robustness will be impacted by how the model weighs new data versus prior data, as reflected by the reduced coefficient of the estimator in the context of models with additional parameters (Appendix A). For example, when calibrating RV217 and CAPRISA 002 (1–2 month) predictions, the estimated intercept is influenced by the later sampling times of the CAPRISA study data (Figure 1 and Appendix A). We are presently preparing a validation of the predictors using sequences from unrelated samples from these and additional cohorts, including data generated using the PACBIO sequencing technology for *env* and *gag* for the CAPRISA 002 cohort. Our ultimate goal is to provide an optimized tool for reproducibly estimating the infection times and founder multiplicities for analysis of the AMP, HVTN 702 and HVTN 705 HIV prevention efficacy studies. Analyzing an additional sequence region (*gag*) will help us determine whether the V3-specific parameters are robust to variations in the underlying effective mutation rate across more- and less-conserved regions of the genome. We note that theoretical arguments and recent unpublished analyses [76] suggest that the star-like phylogeny model employed by Poisson Fitter will be a poor fit after the earliest weeks of infection due to the increased selection pressure resulting from the developing adaptive immune response. For later time points we expect phylogenetic models to provide a better fit, more robust to effects of selection. Our findings on this limited dataset show that after calibration, both estimators can work about equally well at the approximately six-month time point, on par with the calibrated center of bounds estimator, which does not employ any sequence-based estimate. This result reflects the limitations of our method rather than a reflection on the relative appropriateness of the two estimators.

Methods for determining the multiplicity of founders have included aligning and inspecting nucleotide sequences; using a neighbor-joining, maximum-likelihood, and/or Bayesian phylogenetic tree reconstruction methods; and enumerating the number of discernible lineages based on human visual inspection of phylogenetic trees without any use of formal statistical inference [35,47,68,69,77,78,79,80,81,82,83,84]. Our study addresses the need for a robust, reproducible, and automated approach to this particularly important topic toward therapeutic intervention assessment, although bioinformatics expertise is still necessary for, e.g., aligning the input sequences and quality control steps. As shown in Figure 2, the performance of the calibrated predictors of founder multiplicity was independent of the uncalibrated predictor that was included in the model. In fact, even the model using only the covariates (and none of the uncalibrated predictors) yielded these same levels of high classification accuracy, indicating that the driver of the performance of the calibrated multiplicity estimators was the combination of contributions from the other covariates included in these models. The estimates for the parameters that were consistently selected by the LASSO process are shown in Appendix A, refitted to the entire data set without including an uncalibrated predictor. Our study differs from previous efforts to quantify HIV-1 founder multiplicity in that we systematically evaluated multiple procedures to estimate founder multiplicity, including procedures designed to evaluate the hypothesis of a star-like phylogeny as evidence against multiplicity. We also calibrated these methods with respect to various parameters such as neutral mutation rate and strength of selection effects and found that calibration improved the accuracy of these methods.

It remains an open question whether HIV-1 acute infection sequence data are most appropriately modeled by a simple independent-random-variation model lacking any family tree structure (a “star-like” model) or by a phylogenetic model, the latter of which requires the fitting of many more parameters. In the absence of selection, the high mutation and recombination rates of HIV-1 result in rapidly diminishing correlations across position pairs with increased distance between positions. In fact, linkage disequilibrium has been estimated to be around 100 bp in chronic HIV-1 infection [85] meaning that any family tree structure present is limited to regions of about 100 bp. However, evolutionary processes are often modeled well by family tree structures, and environmental changes (i.e., a new drug regimen) can lead to branching phylogenies. Thus, while deviation from the star-like model at timepoints early in infection could signify a multi-founder infection, an alternative explanation is that a branching process model is a better fit.

The absence of source HIV-1 sequence data for comparison with recipient virus is a limitation that exists with most studies to date including this one. Transmitting partner data could increase accuracy and sensitivity in determination of multiplicity and founder identity [84]. Numerous questions remain unanswered about the mechanistic process of HIV-1 infection, including whether the low reported multiplicity of founders is due in part to a within-infected-host competition among potential founders, and what factors affect person-to-person variation in the length of the eclipse phase (after initial exposure to virus, prior to RNA detectability). These important issues will need to be resolved elsewhere. Future studies could attempt to incorporate within-infected-host competition among potential founders for a more comprehensive understanding of the transmitted viral variants and influence of therapeutic interventions. It is also worth noting that the concentration of VRC01 at the site of transmission such as the mucosal tissue may be more relevant to the protective effects in AMP and is an area that needs to be explored further. An additional future avenue of research could incorporate more precise and improved diagnostic bound information by using methods such as Grebe et al. [86] where the qualitative routine diagnostic data can be more precisely evaluated in parallel, compared to the classical Fiebig style staging system currently used in clinical trials. We envision that improvements in establishing infection timing bounds based on these diagnostics will provide improved accuracy in addition to our sequence-based methods. Our study attempts to address an important and globally relevant need to enhance HIV prevention and treatment intervention assessment through improved approaches to estimate timing of HIV infection and founder multiplicity.

## Figures and Tables

**Figure 1 viruses-11-00607-f001:**
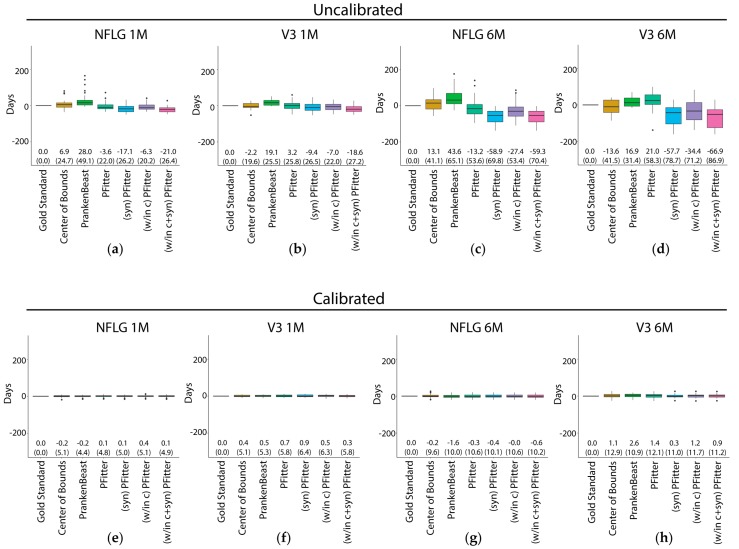
Prediction errors of the Center of Bounds, PrankenBeast, Poisson Fitter, and modified Poisson Fitter estimators of infection time before (**a**–**d**) and after (**e**–**h**) calibration for mutation rate after fitting a linear model with terms for log_10_ plasma viral load (lPVL), the interaction of the estimator with lPVL, the last negative date, the interaction of the estimator with the last negative date and an intercept. Predictions were made on held out data in a leave-on-out cross-validation scheme (see Methods). The sequences used for prediction were: (**a**), (**e**): RV217 (NFLG) 1-2 months; (**b**), (**f**): CAPRISA 002 (V3) 1-2 months; (**c**), (**g**): RV217 ~6 months; (**d**), (**h**): CAPRISA 002 ~6 months. The median difference between the predicted and gold-standard values is shown as the center line of each box; the solid box boundaries illustrate the 25th and 75th percentiles (interquartile range, IQR). The leftmost entry (“Gold standard”) depicts the (zero) “prediction” error if the true days since infection values are known. Values depicted in parentheses indicate the root mean squared error, which is an estimate of the standard error when the fitted predictor is applied to future samples, and the bias is shown above these. The whiskers extend to the most extreme data point within 1.5 times the IQR from the box boundaries; points outside of this range are plotted as outlier points. NFLG, near full-length genome.

**Figure 2 viruses-11-00607-f002:**
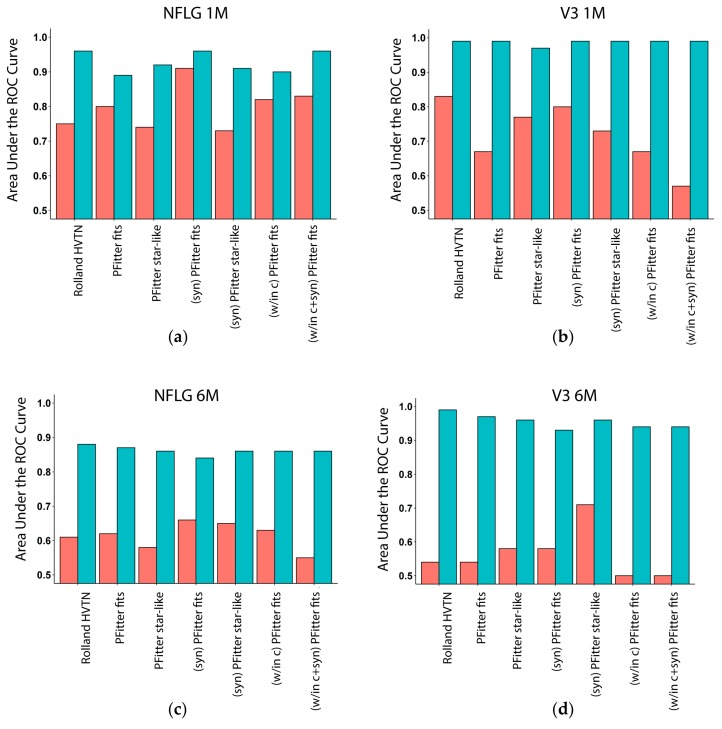
Multiplicity AUC of estimators of multiple-founder infections. Bar plots show areas under the receiver operating characteristic (ROC) curve (AUC) for uncalibrated (red) and calibrated (turquoise) predictors of multiplicity when evaluating predictions on held-out values during leave one-out cross-validation, using the LASSO procedure to select and fit a logistic regression model. Uncalibrated predictors include the method used in the past HVTN sieve analyses, two values computed by the Poisson Fitter software to evaluate a fit to a star-like phylogenetic model, and variants of these using preprocessed inputs (see Methods). Calibrated versions of these predictors are made using models trained using all available data, except for the one participant held out at a time (LOOCV). AUC values of 1.0 indicate a perfect predictor, and values of 0.5 indicate a predictor that is no better than random chance. The sequences used for prediction were: (**a**): RV217 NFLG 1–2 months; (**b**): CAPRISA 002 V3 1–2 months; (**c**): NFLG ~6 months; (**d**): V3 ~6 months.

**Table 1 viruses-11-00607-t001:** Characteristics of HIV-1 sequences and participants in each study.

Study Feature	RV 217 (ECHO)	CAPRISA 002
HIV-1 subtype(s)	CRF01_AE (MSM); A1/D/C and Recombinants (WSM)	C (WSM)
Sequencing strategy	Single genome amplification and sequencing	Next generation sequencing (Illumina w/PrimerID)
HIV-1 genomic region	Near full length genome (NFLG)	V3 variable loop of the gp120 envelope protein
Median bases per HIV-1 sequence (min, IQR, max)	NFLG: 8813 (8624, 8753-8841, 8891); LH:5057 (5027, 5051-5063, 5209); RH:5061 (4898, 5040-5092, 5141)	498 (495, 498-498, 501)
Median HIV-1 sequences per participant after removing recombination and hypermutation (min, IQR, max)	9.5 (2.6, 8.4-10, 11) NFLG: 10 (2, 8-10, 11) LH: 10 (2, 8-10,10) RH: 10 (3, 8-10, 11)	352 (26, 142.3-640, 2764)
Median HIV-1 sequences removed per participant (min, IQR, max)	0 (0, 0-1, 8) NFLG: 0 (0, 0-1.3, 8) LH: 0 (0, 0-0, 4) RH: 0 (0, 0-1, 4)	0 (0, 0-1, 356)
Total number of participants	36	21
Number of MSM	17	0
Number of WSM	19	21
N participants with 1-2M sample	36	20
N participants with ~6M sample	34	18
Mean Gold days 1-2M (SD)	47 (4.3)	62 (4.9)
Mean Gold days ~6M (SD)	184 (11.3)	180 (12.1)
N Gold isMultiple 1-2M (%)	10 (28%)	5 (25%)
N Gold isMultiple ~6M (%)	10 (29%)	6 (33%)
Median bounds width in days 1-2M (min, IQR, max)	48 (20, 34-76, 308)	54 (27, 41-70, 108)
Median bounds width in days ~6M (min, IQR, max)	146 (18, 91-195, 369)	120 (30, 86-170, 183)
Mean lPVL 1-2M (SD)	4.5 (0.8)	4.9 (0.7)
Mean lPVL ~6M (SD)	4.1 (1.0)	4.5 (0.8)

Calculations reflect the evaluated dataset, after removal of sequences with evidence of hypermutation or recombination. NFLG, nearly full length genome; LH, left half genome; RH, right half genome; WSM, women who have sex with men; MSM, men who have sex with men; IQR, interquartile range; lPVL = log_10_ plasma viral load; SD, standard deviation; Gold = modified center of bounds (COB) timing estimate applied to previously unavailable acute tight diagnostic bounds (prior to the 1-2M sample date) and the agreed-upon gold standard is a multiple indicator, see Methods.

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
