# Peer review of "Combining Viral Genetics and Statistical Modeling to Improve HIV-1 Time-of-Infection Estimation towards Enhanced Vaccine Efficacy Assessment"

_viruses, 2019, doi:10.3390/v11070607_

Round 1

Reviewer 1 Report

In this manuscript, Rossenkhan evaluated various strategies (novel and existing) to determine the time of HIV-1 infection and multiplicity of HIV-1 variants. Since this information most often is missing due to lack of very early samples, improved or new methods for estimation of these important parameters are warranted. 

Overall, the manuscript is well-written and outlines the study in a logical and easy way. However, the manuscript is very meticulous and detailed which clearly show the authors' high ambitions with the study. This is admirable and very helpful for initiated readers with an interest in this topic. Unfortunately, this has resulted in a very long manuscript and there is a risk that readers that are less interested in the specific details of the various methods used may lose interest and focus. The main take-home messages also get drowned a bit in the wealth of data and details. My main comment is therefore that the authors consider to restructure and shorten the manuscript and make it more concise. A large proportion of the methods section could be moved to supplementary material.

Specific comments:

1. The manuscript contains a large proportion of self-citations. Both the AHI-field and the field of sequence analysis as a mean to determine infection date are very large, and many researchers have made important contributions over the years. The authors should reassess the literature and add additional key articles.

2.  A broader discussion about the similarities and differences between RV217 and CAPRISA 002 and how these can have influenced the results are needed. For example, differences in infecting subtype (how generalizable are the results across subtype and CRFs?), transmission routes (are the results generalizable to the general population and/or other rsik groups?), females vs. males, ethnicities, etc. Are there any caveats with the viral loads, how they were determined, the number of measurements etc.

3. Some of the sections seem to be miss-referenced, e.g. "2.7 PrankenBeast". The authors should carefully go through the reference lost for accuracy.

4. Some sections contain several strong statements that are not supported by proper statistics and p-values, e.g. paragraph 3.4; "The estimators showed similar performance..." "...the PrankenBeast estimates were much better...". The authors need support all strong statements with proper statistical tests and wording.

Author Response

Reviewer: 1
Comments to the Author
In this manuscript, Rossenkhan evaluated various strategies (novel and existing) to determine the time of HIV-1 infection and multiplicity of HIV-1 variants. Since this information most often is missing due to lack of very early samples, improved or new methods for estimation of these important parameters are warranted. 

Overall, the manuscript is well-written and outlines the study in a logical and easy way. However, the manuscript is very meticulous and detailed which clearly show the authors' high ambitions with the study. This is admirable and very helpful for initiated readers with an interest in this topic. Unfortunately, this has resulted in a very long manuscript and there is a risk that readers that are less interested in the specific details of the various methods used may lose interest and focus. The main take-home messages also get drowned a bit in the wealth of data and details. My main comment is therefore that the authors consider to restructure and shorten the manuscript and make it more concise. A large proportion of the methods section could be moved to supplementary material.

Reviewer 1_response overall:

We appreciate this valuable point brought up by the reviewer; the manuscript has been substantially re-structured and shortened as per reviewer suggestion. We however kindly request to keep the current version of our methods section intact for a more comprehensive and complete understanding of our work. Multiple sections have been moved to the supplementary material to reduce verbosity, and we endeavored to reduce verbosity in the main manuscript text.

Specific comments:

1. The manuscript contains a large proportion of self-citations. Both the AHI-field and the field of sequence analysis as a mean to determine infection date are very large, and many researchers have made important contributions over the years. The authors should reassess the literature and add additional key articles.

Reviewer 1_response specific comment 1:

We appreciate the Reviewer's comment and have updated citations as per reviewer suggestion. We acknowledge that a wealth of literature exists, but we have restricted our citations to those mainly addressing day-of-infection timing in acute infection studies as they are the main scope of this study. Moreover, there is a substantial literature on estimation of acute infection based on “cross-sectional prevalence testing”, which seeks to estimate acute infection with resolution at the level of many weeks or months.  This literature is not as directly relevant to our objective to pinpoint infection dates within days or a small number of weeks based on frequent longitudinal HIV testing, which explains why we include relatively few references from the cross-sectional prevalence literature. If we have missed any specific and pertinent additional key articles, we would greatly appreciate suggestions.

2.  A broader discussion about the similarities and differences between RV217 and CAPRISA 002 and how these can have influenced the results are needed. For example, differences in infecting subtype (how generalizable are the results across subtype and CRFs?), transmission routes (are the results generalizable to the general population and/or other risk groups?), females vs. males, ethnicities, etc. Are there any caveats with the viral loads, how they were determined, the number of measurements etc.

Reviewer 1_response specific comment 2:

We appreciate this point. We have added a paragraph addressing similarities and differences between RV217 and CAPRISA 002 (Page 16, Lines 687-698). With regards to generalizability to other subtypes it is difficult to know, given ours is the first study utilizing very acute infection sequences to evaluate HIV-1 infection timing methods with approximately known true infection dates, and for this study we are limited to the RV217 and CAPRISA 002 data subsets described in Table 1, which include study samples with specific study characteristics (gender, subtype etc.). A more extensive assessment of robustness of these findings and calibrated models is not possible at this time due to the fact that these types of datasets are rare. As described below, we have added text to explicitly call the reader’s attention to the requirement for further validation.

3. Some of the sections seem to be miss-referenced, e.g. "2.7 PrankenBeast". The authors should carefully go through the reference lost for accuracy.

Reviewer 1_response specific comment 3:

We appreciate the reviewer drawing our attention to this, and have double checked our citations and references.  This url is correct; Erick Matsen uses the names PrankenBeast and PREAST interchangeably and has now clarified this on the github page. Thank you for pointing out that the url appears inconsistent. We have clarified in the paper that this method is also known as PREAST. Note that the referenced section is now section 2.8 (page 7, line  302).

4. Some sections contain several strong statements that are not supported by proper statistics and p-values, e.g. paragraph 3.4; "The estimators showed similar performance..." "...the PrankenBeast estimates were much better...". The authors need support all strong statements with proper statistical tests and wording.

Reviewer 1_response specific comment 4:

We thank the reviewer for this comment, which is well-taken. We have corrected all such statements to clarify the message and to clearly indicate that they are not meant to reflect a formal comparison or confidence statement (e.g., page 11, lines 501-504). We are only presenting summary statistics of performance of predictors; we are not presenting uncertainty intervals and we have not conducted formal comparisons for the bias and RMSE, nor for the AUC-ROC values. The uncertainty in these values due to variation across similarly-sized samples from the larger populations that they represent is potentially estimable through comparisons across similarly-sized samples, or through resampling-based methods, eg. using the bootstrap. Estimations of the sampling variation in these values is beyond the scope of our present abilities given these limited data, so we restrict our comments to the observed samples, and make no claims about population parameters.

Reviewer 2 Report

The study aims to improve measures of time since HIV infection for vaccine efficacy studies, and more generally for other situations where multiple proxy measures are available, including time of first positive test, viral load, and sequence diversity metrics. A key premise is that incorporating multiple measures improves the precision of estimates. This seems uncontroversial, as each metric has been previously shown to correlate to some extent with time since infection, and it is encouraging that including all of them in a single more complex model reduces the variance of the predictions. Indeed, this is the basis of the current MAA (mult-assay algorithms) used for recency testing in many public health laboratories. However, given the cost of viral load assays and deep sequencing, the cost of multi-assay methodologies is prohibitive in many settings.

Overall, the paper is extremely verbose and unfortunately difficult to break down into the significant parts – aims, hypotheses, how each method was used to address these, and clear results for each aim. After reading the paper closely I was still uncertain of key elements of the results:

(1)  Which metrics were the most significant in the final model, and how much of the variance did they explain? Was this the same for both datasets?

(2)  The model was trained on all available data, with error estimates made on a leave-one-group-out basis, where groups were sequences from the same participant. This is fine in principle, but does not demonstrate true performance of the model on a held-out (unseen) dataset, particularly because features of the data here are very likely to create correlations between observations that are not due to the shared participant ID (for example, the nature of the sequencing protocol). In practice, the true performance of this model may be substantially worse than the validation step suggests. If it is impossible to have an entirely clean held-out set, I would prefer to see grouped k-fold validation, rather than leave-one-out.

(3)  What data types are required for this model to work? It sounds as though viral load measurements, dates of diagnosis and deep sequencing (or multiple sequences per participant) are necessary. If this is the case, it’s a huge ask -- how sensitive is the model to missing data? 

(4)  More specifically, if the date of diagnosis is critical to the model, how much tolerance is permissible in cases where the date is wrong, or missing? Re-testing is common in many settings. This date will in practice often be the sampling date. 

(5)  What is the purpose of Figures 1 and 3 – it seems intuitive that approximate rather than precise measures of time since infection would reduce power, and it does not directly address the aims of the paper to demonstrate the extent to which this happens. I was also puzzled by the idea of generating pseudo-COB intervals around the Gold-standard times since infection – why was a uniform distribution used? Is the purpose of this first section purely theoretical? It feels like it belongs in a separate paper. There is a mysterious reference in the methods to unpublished results on the same subject, and it was not clear to me how these differ from Figure 1.

(6)  A comment specifically on the software pipeline: if the software is to be made available to the community, it would be best to provide a Docker container or similar, containing all the (many) dependencies. It should be made clear whether the authors advocate the use of their specific software (in which case efforts should be made to make it available in simple form to the reviewers, with test data), or whether they are describing a method only.  

Estimating time since HIV infection is a notoriously difficult problem and the authors do a thorough job of exploring available metrics. The explanation of the background and methods is interesting, but extremely verbose and does little to help the reader understand the significance of the findings. Many common terms are unnecessarily re-named (using “family tree” and “branching-process model” instead of “phylogeny” for example). I would like to see this work presented in a more focused way, making it clear what specific aspect of recency estimation is being addressed, how (or whether) the reader is expected to use the tools, and what the true error rates are on a new dataset that was not at *any* stage used during training the model. There should also be some discussion of the practicality of implementing this approach in the place of standard MAA recency testing, and what the benefits of this would be in the chosen setting (eg. vaccine efficacy trials).

Author Response

Reviewer: 2

The study aims to improve measures of time since HIV infection for vaccine efficacy studies, and more generally for other situations where multiple proxy measures are available, including time of first positive test, viral load, and sequence diversity metrics. A key premise is that incorporating multiple measures improves the precision of estimates. This seems uncontroversial, as each metric has been previously shown to correlate to some extent with time since infection, and it is encouraging that including all of them in a single more complex model reduces the variance of the predictions. Indeed, this is the basis of the current MAA (mult-assay algorithms) used for recency testing in many public health laboratories. However, given the cost of viral load assays and deep sequencing, the cost of multi-assay methodologies is prohibitive in many settings.

Overall, the paper is extremely verbose and unfortunately difficult to break down into the significant parts – aims, hypotheses, how each method was used to address these, and clear results for each aim. After reading the paper closely I was still uncertain of key elements of the results:

(1)  Which metrics were the most significant in the final model, and how much of the variance did they explain? Was this the same for both datasets?

Reviewer 2_response specific comment 1

We appreciate the reviewer’s feedback that this information is of interest. For timing estimation, for the model that includes the estimate from PFitter, the viral load, and the lower bound on the time since infection from diagnostic tests, the plasma viral load contributes the most information. On Lines 672-685 there is a discussion of R2 values for this timing calibration model (overall less than 0.20) versus a model employing lPVL alone (0.80, indicating that this viral load model alone would explain 80% of the variation). We did not evaluate a model containing only the diagnostic bound. For the multiplicity results we have now added a new column to Supplementary Table 4 showing the percent of variance explained, calculated as the square of the Pearson correlation coefficient between the predictor and the gold standard. The most informative input to the multivariate calibration model was the pairwise sequence diversity, which inflates when there are multiple founders. Interestingly, the output of DSStarPhyTest contributed to the model through the difference between evidence of a fit when applying the method to unaltered inputs (“DS StarPhy R”) versus to the data after masking nonsynonymous codons and then clustering the sequences (“(w/in clusts) (syn) DS StarPhy R”).  This is sensible, because in the case of a multi-founder infection, clustering the sequences first and computing only within-cluster fits should improve the estimate. As discussed in Methods (lines 256-360), DS StarPhyTest is a reimplementation of PFitter’s fits statistic  that uses Dempster-Shafer Analysis to compute a measure of the goodness of fit of the star-like phylogenetic model. We have added text to the Results section (lines 623-630) to address the contributions of these variables to the models.

(2)  The model was trained on all available data, with error estimates made on a leave-one-group-out basis, where groups were sequences from the same participant. This is fine in principle, but does not demonstrate true performance of the model on a held-out (unseen) dataset, particularly because features of the data here are very likely to create correlations between observations that are not due to the shared participant ID (for example, the nature of the sequencing protocol). In practice, the true performance of this model may be substantially worse than the validation step suggests. If it is impossible to have an entirely clean held-out set, I would prefer to see grouped k-fold validation, rather than leave-one-out.

Reviewer 2_response specific comment 2

We agree with reviewer 2 that in an ideal situation a completely held out dataset, or short of that, k-fold validation, would be good choices. There is a rich ongoing debate among machine learning practitioners about this topic, and there are several good reasons to consider k-fold and other variants of cross-validation over LOOCV. However, the tradeoff of bias and variance is not a simple way to resolve the correct choice of k (see for instance https://stats.stackexchange.com/questions/90902/why-is-leave-one-out-cross-validation-loocv-variance-about-the-mean-estimate-f), and in practice the reduced variance of a smaller choice for k may not be worth the increased bias. Due to the fact that these types of data sets are very rare, we are limited to a small sample size in each group. In our experience this provides a reasonable justification for employing LOOCV, since even with k = n, the sizes of the retained datasets approach the limits of applicability of asymptotic-based procedures such as the standard and logistic regression models that we have employed for calibration. We have added the statement, “The choice of leave-one-out (versus k-fold) cross-validation was made because of the small sample sizes of these datasets (Table 1; n = 18 to 36)” Page 9, Line 374-380. We also added a note about this in the conclusion of the Results section (lines 670-672, reading, “The structure of these data constrained our ability to employ alternative forms of cross-validation, especially relatively small sample sizes within each time / region dataset (Table 1).”

We also concur that separate datasets, or doubly nested cross-validation, would be needed to obtain fully honest estimates of prediction error generalizing to new samples.  Given that it does not appear to be possible to do this with the available data, we have added stronger caveats in multiple locations in the paper, including in the Abstract (lines 47 to 49) reading, “These results have not yet been validated on external data, and the best-fitting models are likely to be less robust than simpler models to variation in sequencing conditions.” In the conclusion of the Results section we added, “However, further validation is needed to evaluate the robustness of these specific calibration model fits when applied to different sequencing technologies, HIV-1 subtypes, and sampling times. We recommend further validation with data more similar to the target setting, and potentially further calibration, before employing these calibrated models in practice.” (lines 694-698), and to the Discussion (lines 745-748), “The two datasets we analyzed were not large enough to facilitate fully honest estimates of prediction error generalizing for new samples; therefore when additional acute infection datasets become available it will be important to conduct such analyses.”

(3)  What data types are required for this model to work? It sounds as though viral load measurements, dates of diagnosis and deep sequencing (or multiple sequences per participant) are necessary. If this is the case, it’s a huge ask -- how sensitive is the model to missing data? 

Reviewer 2_response specific comment 3

We thank the reviewer for pointing out where we need to be clearer about the requirements and the context. First we note that it is presently essential to obtain multiple sequences per participant when employing these sequence-based estimators of time since infection. To our knowledge there is no existing methodology for employing bulk/consensus sequencing in acute infection with the goal of estimating the time since infection. The inclusion of viral loads and diagnosis dates improves model performance in our evaluation, although we also provide results of employing an estimate-only model lacking the plasma viral load information (Supplementary Table 3), and have models built into the software for contexts in which both diagnostic-based bounds and plasma viral load are unavailable. The context of our study is the analysis of incident infections in HIV-1 prevention efficacy trials; we expect that these data would be available in this context. We have added the clarification statement Page 2 Lines 82-88 reading, “The focus of our study is to evaluate and optimize day-of infection estimators using data available from a single sample collected in a time frame reflecting diagnosis intervals typical of HIV-1 prevention efficacy trials: 1-2 months, or approximately 6 months, post infection. These data include aligned HIV-1 nucleotide sequences and plasma viral load from the first HIV-positive sample; in addition, the dates and results of HIV-1 diagnostic tests for at least the last negative and first positive diagnosis dates are available for all infected participants.”

(4)  More specifically, if the date of diagnosis is critical to the model, how much tolerance is permissible in cases where the date is wrong, or missing? Re-testing is common in many settings. This date will in practice often be the sampling date. 

Reviewer 2_response specific comment 4

As noted above and now made clearer in the manuscript, our goal is to improve the estimate of the time of infection in the context of clinical trials where HIV testing is done at a series of longitudinal testing visits (e.g., 4-weekly in AMP for 80 weeks) and diagnosis of HIV infection is the primary endpoint of the trial, which involves confirmatory testing for all endpoints. We generally obtain reasonably reliable data on the date of diagnosis within a fairly narrow time window.  However, some participants miss study visits, so the diagnosis date may happen substantially after the last negative sample date. In rare cases where the diagnosis date is wrong or missing, we are limited. However, by sampling actual intervals between visits from MTN and HVTN clinical trials in the employment of the diagnostic bounds for the study, the rate at which actual visits are missed is reasonably accommodated into our findings. We have added a section to the Methods to more clearly explain the procedure for sampling these bounds (new section 2.4, lines 230-247), and we have added text to the Results section to highlight the right skew in the histogram of lengths of these intervals that was induced by missed visits in the source trial (HVTN 502 or MTN 003): (line 456-459) “Note that these interval lengths have distributions that are observed in actual clinical trials, and as such they include some right skew and high outliers reflecting that some infected participants are missing one or more of their scheduled visits for diagnostic testing (Table 1).” Our impression in exploratory analysis of models lacking diagnostic bounds is that they perform much more poorly than models that do include them, even given the wide intervals that the bounds often convey.

(5)  What is the purpose of Figures 1 and 3 – it seems intuitive that approximate rather than precise measures of time since infection would reduce power, and it does not directly address the aims of the paper to demonstrate the extent to which this happens. I was also puzzled by the idea of generatin pseudo-COB intervals around the Gold-standard times since infection – why was a uniform distribution used? Is the purpose of this first section purely theoretical? It feels like it belongs in a separate paper. There is a mysterious reference in the methods to unpublished results on the same subject, and it was not clear to me how these differ from Figure 1.

Reviewer 2_response specific comment 5

Our intent in including Figure 1 was to highlight the importance of improving estimates of HIV infection time on vaccine efficacy assessments, as motivation for our research. We however agree with Reviewer 2 that this may distract from the main topic of our manuscript and have moved Figure 1 (now supplementary Figure 1) and associated text to the supplementary material.  We have additionally moved Figure 2 (now Supplementary Figure 2) with associated text to the supplementary section to reduce the highlighted verbosity concern. In addition, we have moved Figure 3 to the supplementary section (now Supplementary Figure 3), in further efforts to address reviewer concerns regarding flow of the manuscript and verbosity. As noted above, we have also clarified the use of true and artificial bounds with an explanation of the distribution in the new section 2.4 to address the reviewer’s concern.

(6)  A comment specifically on the software pipeline: if the software is to be made available to the community, it would be best to provide a Docker container or similar, containing all the (many) dependencies. It should be made clear whether the authors advocate the use of their specific software (in which case efforts should be made to make it available in simple form to the reviewers, with test data), or whether they are describing a method only.   

Estimating time since HIV infection is a notoriously difficult problem and the authors do a thorough job of exploring available metrics. The explanation of the background and methods is interesting, but extremely verbose and does little to help the reader understand the significance of the findings. Many common terms are unnecessarily re-named (using “family tree” and “branching-process model” instead of “phylogeny” for example). I would like to see this work presented in a more focused way, making it clear what specific aspect of recency estimation is being addressed, how (or whether) the reader is expected to use the tools, and what the true error rates are on a new dataset that was not at *any* stage used during training the model. There should also be some discussion of the practicality of implementing this approach in the place of standard MAA recency testing, and what the benefits of this would be in the chosen setting (eg. vaccine efficacy trials).

Reviewer 2_response specific comment 6

We thank the reviewer for this very helpful suggestion regarding Docker. A Docker image has been created and is available from Docker Hub at https://hub.docker.com/r/philliplab/hiv-founder-id. See the description on the repository for instructions and information about the test data. We additionally address the concern related to terminology throughout the manuscript and have made significant efforts to present the work in a more focused way as per the reviewer suggestion. Finally, we added material to section 3.4 (Calibration, considerations and results summary) to address the reviewer concern about correctly estimating true error rates.

Round 2

Reviewer 2 Report

The authors are to be commended for making substantial revisions to the manuscript, and for detailed responses to specific comments by both reviewers. In particular, the restriction of the scope of the paper has improved it substantially. Some remaining points for minor revision:

(1) Please update the title to reflect that the time-of-infection in question refers only to extremely recent HIV infections. 

(2) The new text in lines 272 - 276 is difficult to interpret - please rephrase/clarify the use of "longitudinal" and "expected time of initial sequencing". 

(3) New text on software pipeline in section 2.14 - please move most of this to supplementary material and leave only the link to the docker container and the note about application to new data. I am not able to test the docker container example here due to time constraints, so cannot verify this is functional. Please ensure that at least one person outside your immediate computational infrastructure has tried to download, install, and run the example, and *provide the name(s)* of all testers (and include them in acknowledgements).

(4) Please go through the text again to remove redundancies - it is still verbose and very long. Example: "However, while around 80% of all acute, sexually-transmitted HIV infections are established by a single founder lineage, about 20% are established by multiple founder lineages" -- sentences like this are unnecessarily long; it is just as clear to say that "about 20% if HIV infection is established by multiple founder viruses" (because obviously all infections must add up to 100%). I do not have time to pick out all such examples but there are many.

Author Response

Reviewer(s)' Comments to Author:

Reviewer:

The authors are to be commended for making substantial revisions to the manuscript, and for detailed responses to specific comments by both reviewers. In particular, the restriction of the scope of the paper has improved it substantially. Some remaining points for minor revision:

(1) Please update the title to reflect that the time-of-infection in question refers only to extremely recent HIV infections. 

Response to reviewer:

We thank the reviewer and have updated the title to be more reflective of this “Combining Viral Genetics and Statistical Modeling to Improve HIV-1 Time-of-infection Estimation Towards Enhanced Vaccine Efficacy Assessment”. Additionally, we have added the keywords “acute and early HIV-1 infection” to update the context of our study for improved searchability.

(2) The new text in lines 272 - 276 is difficult to interpret - please rephrase/clarify the use of "longitudinal" and "expected time of initial sequencing". 

Response to reviewer:

We thank the reviewer for their comment and suspect there might have been a typo in the referenced line numbers (as 272-276 do not make reference to the cited phrases “longitudinal” and “expected time of initial sampling”). We have made the suggested rephrasing/clarification to what we think is being referred to (section 2.4 lines 621-638) and hope that these changes convey the message in a more easily interpretable way as per reviewer suggestion.

(3) New text on software pipeline in section 2.14 - please move most of this to supplementary material and leave only the link to the docker container and the note about application to new data. I am not able to test the docker container example here due to time constraints, so cannot verify this is functional. Please ensure that at least one person outside your immediate computational infrastructure has tried to download, install, and run the example, and *provide the name(s)* of all testers (and include them in acknowledgements).

Response to reviewer:

Thank you for the helpful recommendation, we have moved the text to the supplementary section and requested that two researchers outside our immediate computational infrastructure verify functionality. Art Poon and Natasha Wood kindly provided valuable input, edits have been made to the docker image and their contribution has been added to the acknowledgement section.

(4) Please go through the text again to remove redundancies - it is still verbose and very long. Example: "However, while around 80% of all acute, sexually-transmitted HIV infections are established by a single founder lineage, about 20% are established by multiple founder lineages" -- sentences like this are unnecessarily long; it is just as clear to say that "about 20% if HIV infection is established by multiple founder viruses" (because obviously all infections must add up to 100%). I do not have time to pick out all such examples but there are many.

Response to reviewer:

We have edited the sentence mentioned by the reviewer and have gone through the text to remove other similar redundancies. We would once again like to sincerely thank and express our gratitude to the reviewers for their time, meticulous critique and recommendations as these have improved the flow and research we wish to communicate.